# Translational Control using an Expanded Genetic Code

**DOI:** 10.3390/ijms20040887

**Published:** 2019-02-18

**Authors:** Yusuke Kato

**Affiliations:** Division of Biotechnology, Institute of Agrobiological Sciences, National Agriculture and Food Research Organization (NARO), Oowashi 1-2, Tsukuba, Ibaraki 305-8634, Japan; kato@affrc.go.jp; Tel.: +81-29-838-6059

**Keywords:** genetic switch, synthetic biology, unnatural amino acids, translational regulation, biological containment, stop codon read-through

## Abstract

A bio-orthogonal and unnatural substance, such as an unnatural amino acid (Uaa), is an ideal regulator to control target gene expression in a synthetic gene circuit. Genetic code expansion technology has achieved Uaa incorporation into ribosomal synthesized proteins in vivo at specific sites designated by UAG stop codons. This site-specific Uaa incorporation can be used as a controller of target gene expression at the translational level by conditional read-through of internal UAG stop codons. Recent advances in optimization of site-specific Uaa incorporation for translational regulation have enabled more precise control over a wide range of novel important applications, such as Uaa-auxotrophy-based biological containment, live-attenuated vaccine, and high-yield zero-leakage expression systems, in which Uaa translational control is exclusively used as an essential genetic element. This review summarizes the history and recent advance of the translational control by conditional stop codon read-through, especially focusing on the methods using the site-specific Uaa incorporation.

## 1. Introduction

Conditional induction of gene expression is one of the most important technologies for constructing synthetic gene circuits, and such regulation has applications from basic research to industrial use. Gene expression can be controlled at various levels, such as transcription, post-transcription, translation, and post-translation. Among these, transcriptional control is most frequently used because many genetic parts, such as inducible promoters and repressors, have been established. On the other hand, there are only a limited number of methods for control at the translational level.

An established method for translational control is the conditional stop codon read-through (Figure 1). Three codons, which do not encode for any amino acids, UAG (amber), UAA (ochre), and UGA (opal), are defined as stop codons. During peptide synthesis on the ribosome, stop codons are recognized by peptide-release factors, resulting in termination of elongation and release of the peptide from the ribosome. Insertion of a stop codon, usually UAG for synthetic gene circuits, in a protein coding region of a target gene causes production of an unusual truncated protein in which protein synthesis is terminated at the stop codon. The target gene is not expressed if the truncated protein is not functional. However, a mutant tRNA that recognizes the inserted stop codon (stop codon suppressor tRNA) can rescue the function of the target gene product by incorporation of an amino acid at the inserted stop codon (stop codon read-through), which results in a production of the full-length protein. Therefore, the expression of a target gene can be controlled at the translational level by conditional production of an aminoacylated stop codon suppressor tRNA.

In this review, we describe the control of target gene expression in vivo by conditional stop codon read-through, especially focusing on the recently advanced method using a site-specific unnatural amino acid (Uaa) incorporation system. This review does not cover a drug-stimulated translational read-through of stop codons [1].

## 2. Classifications

We define three categories, termed here as the first to third generation (Figure 2). The first generation (G1) method uses a UAG suppressor tRNA (tRNA_CUA_) carrying a standard amino acid. The second generation (G2) method involves application of a site-specific Uaa incorporation system. This system uses a mutant pair of orthogonal aminoacyl-tRNA synthetase (aaRS)/tRNA_CUA_, which specifically incorporates the Uaa at the UAG codon. The third generation (G3) is an improved method of G2 with various genetic modifications to optimize G2 for switching use. In the following section, the features of each category are described in detail.

## 3. G1

G1 is defined as a genetic switch which uses a tRNA_CUA_ carrying a standard amino acid to suppress the UAG stop codon (Figure 2). The translation of target gene mRNAs can be controlled by a conditional generation of aminoacyl-tRNA_CUA_.

A mutation of a sense codon, which locates in the protein coding region to a stop codon (nonsense mutation), often causes a loss-of-function of the protein. Surprisingly, such a finding was already reported before the establishment of a complete codon table [2,3,4]. The loss-of-function caused by a nonsense mutation can mostly be rescued by a nonsense suppressor tRNA [2,3,5]. The nonsense suppressor tRNA is mainly generated from near-cognate tRNAs by a mutation at least in the anticodon, and it recognizes the stop codon [6]. tRNA_CUA_ recognizes the UAG stop codon and incorporates an amino acid. For example, *Eschrichia coli supE* (*Su-2*) is a tRNA_CUA_ gene derived from the tRNA^Gln^ gene (*GlnV and GlnX*, tRNA_CUG_) [7]. The *supE* tRNA_CUA_ incorporates Gln at UAG [8]. Synthesis of the full-length protein is recovered by an aminoacylated-tRNA_CUA_ (UAG read-through), resulting in reversion of the TAG interrupted gene to a functional gene.

G1 controls the translation of target gene products using the UAG read-through. In this method, TAG(s) is inserted in the protein coding region of the target gene. Only a three nucleotide insertion (TAG) or point mutation(s) is needed for TAG insertion. For example, a point mutation of CAG which encodes Gln generates TAG. Such mutations can also be isolated from a natural population. G1 is widely used for functional analysis of specific genes by conditional rescue of either a naturally occurring or a synthetic TAG-insertion into mutant genes [9,10]. The TAG-insertion in a target gene sequence may be technically simple and easy, suggesting that this is an advantage of the translational switches, including both G1 and the following generation methods.

On the other hand, a limitation of the translational control using UAG read-through is its off-target effects. The aminoacyl-tRNA_CUA_ causes UAG read-through not only at the UAGs which are artificially inserted in the target genes, but also at the native UAGs encoded in the genome. The suppression of genomically encoded UAGs may occasionally cause undesirable effects. Although most laboratory strains of *E. coli* are tolerant of UAG read-through, growth of the DH10B strain was strongly inhibited, suggesting a species-specific toxicity [11]. 

UAG read-through is controlled by a conditional production of aminoacyl-tRNA_CUA_. Some control methods have been reported, as described in the following sub-sections.

### 3.1. Thermolabile tRNA_CUA_

A temperature-sensitive mutant of tRNA_CUA_ can be used for induction of UAG read-through (Figure 3A) [9,12,13,14,15,16,17]. An *E. coli* strain carrying the gene of a temperature-sensitive tRNA_CUA_, supF, incorporates Phe at the UAG codon at 30 °C [9]. In contrast, the supF tRNA_CUA_ is inactive at 42 °C. The UAG read-through, therefore, can be controlled by selection of the culture temperature.

### 3.2. Inducible Transcription of tRNA_CUA_

Conditional transcription of tRNA_CUA_ is a direct and effective method (Figure 3B) [18,19,20]. Inducible transcriptional controllers, such as the transcriptional regulatory elements of araBAD operon (araO/araC) and the tetracycline operator-repressor system (tetO/tetR), were used for the induction of tRNA_CUA_. 

### 3.3. Inducible Aminoacyl-tRNA Synthetase

The production of aminoacyl-tRNA_CUA_ can also be controlled by conditional expression of the cognate aaRS, which charges an amino acid onto the tRNA_CUA_ (Figure 3C) [21]. This method is available only for an orthogonal aaRS/tRNA_CUA_ pair in which the tRNA_CUA_ is exclusively aminoacylated by its cognate aaRS, but not recognized by the host aaRS’s. Such orthogonal aaRS/tRNA pairs are usually isolated from evolutionarily distant organisms [22]. *E. coli* GlnRS/tRNA^Gln^ pair is orthogonal in mammalian cells [23]. Conditional UAG read-through was achieved using an inducible *E. coli* GlnRS with its cognate UAG suppressor tRNA^Gln^ [21].

### 3.4. Introduction of the tRNA_CUA_ Gene

Direct introduction of a tRNA_CUA_ expression construct is also a method for a temporal UAG read-through (Figure 3D). For example, protoplasts, which were electroporated with the tRNA_CUA_ gene, showed effective UAG read-through by a factor of at least several hundred-fold [24]. Genetic introduction is an alternative method. Toxin-mediated ablation of a specific cell population was achieved both in animals and plants using a genetically conditional tRNA_CUA_ expression [25,26]. This ablation used a transgenic line carrying a UAG-inserted toxin gene after crossing with a strain harboring a tRNA_CUA_.

## 4. G2

G2 uses an orthogonal engineered aaRS/tRNA_CUA_ pair, which is specific for the Uaa (Figure 2). UAG read-through is induced in the presence of the Uaa. 

Some Uaa’s whose structures are similar to standard amino acids can be incorporated in ribosomal synthesized proteins [27,28,29,30,31]. This is possible due to the imperfect substrate specificity of aaRS, which mischarges the Uaa to its cognate tRNA. The finding of Uaa incorporation was surprisingly early and occurred before the discovery of aaRS and tRNA [32,33,34]. Using this method, a protein which contained a fluorine-labeled amino acid was prepared for a structural analysis using ^19^F NMR [35,36,37,38,39]. Moreover, a protein substituted with Uaa’s, referred to as an “alloprotein”, was proposed for designing rare functions of specific proteins [40]. A conformation-stabilized human epidermal growth factor was generated as a proof-of-concept study of an alloprotein. The usefulness of these applications stimulated further progress on the Uaa incorporation technology. A tRNA_CUA_, which was chemically aminoacylated with an Uaa, was used to incorporate the Uaa at a specific site which was designated by UAG in a ribosomal synthesized protein in vitro [41]. In the early 2000′s, an orthogonal tRNA_CUA_ was successfully aminoacylated using a Uaa-specific mutant of aaRS (UaaRS), enabling use of this method in vivo. [42,43,44,45,46,47]. This method is called a site-specific Uaa incorporation.

G2 is a method to control the translation of target genes using the site-specific Uaa incorporation system [48]. The UaaRS/tRNA_CUA_ genes is introduced in a host. The TAG(s) is inserted in the target genes, in a similar manner to that of G1. The Uaa is added to the extracellular space and is transported into the intracellular space. Subsequently, the Uaa is incorporated into the target gene products at the inserted UAG(s). In contrast, translation of the target gene products is terminated at the UAG in the absence of the Uaa, resulting in no functional proteins being produced.

The most remarkable advantage of G2 is that the regulatory molecule is a Uaa [48]. Uaa is a synthetic molecule and does not exist either in the natural environment or in living organisms, suggesting that an unexpected turning on of G2 by an environmental or an endogenous Uaa is not a possibility. In addition to the all-or-none switching, G2 can control the translational efficiency of target gene products at any intermediate magnitude by adjustment of the Uaa concentration [49]. This tunability is observed in an individual cell and is not due to the change of population average of the induced and non-induced cells. The switching of G2 is reversible [50]. Addition or removal of the Uaa can induce OFF-ON and ON-OFF transition, respectively. Such transitions are completed within about 20–30 min in *E. coli*.

In conclusion, G2 is an excellent translational controller, which is environmental and endogenous noise-free, tunable, fast responding, and reversible. However, some unique problems have been found with G2, as described in detail in the next section. 

## 5. G3

G3 is an advanced/improved version of G2, which is optimized for switching purposes (Figure 2). G2 uses a site-specific Uaa incorporation system without any modification. However, this system was originally developed to alter target proteins for various purposes, including structural analysis, labeling, and functional modification of proteins [51,52,53]. Therefore, G2 has not been sufficiently optimized for use as a genetic switch. This problem has been solved with the G3 methodology.

A good genetic switch should satisfy two fundamental requirements. First, the yield should be as high as possible. Here, the yield is defined as the magnitude of protein production under the induced condition (ON-state). In the Uaa translational switch, the ON-state means that the optimized concentration of the Uaa is supplied. The dynamic range of protein production is restricted by the yield. The UAG read-through competes with the termination of peptide elongation by a peptide release factor at the UAGs. The efficiency is the ratio of the full-length protein, which is produced by UAG read-through, to a truncated protein, which is produced by the termination of peptide elongation at the UAG. A higher efficiency should achieve a higher yield. In contrast, the shortage of UaaRS/tRNA_CUA_ may cause a lower yield.

Second, the leakage should be as low as possible. The leakage is the protein produced under the suppression condition (OFF-state). In the Uaa translational switch, the OFF-state means that the Uaa is absent. Leakage of the Uaa translational switch consists of two components [50] (Figure 4). One component is the Leakage Translation attributable to the Uaa translational Switch (LTaS). The LTaS is mainly caused by either a natural amino acid mischarge to tRNA_CUA_ by an incompletely specific UaaRS or a host aaRS, suggesting that the substrate specificity of UaaRS and the orthogonality of tRNA_CUA_ affect the LTaS level [48]. Another component is the Leakage Translation attributable to the Basal Read-through (LTaBR). The basal read-through by near cognate tRNAs causes the LTaBR [54,55,56]. This component is independent from the performance of the Uaa translational switch. Both LTaS and LTaBR, therefore, must be suppressed to achieve the lowest leakage translation.

The fundamental performance of a genetic switch, how high the yield is, and how low the leakage is, is often called “tightness” and is evaluated using the ratio between the yield and leakage. Here, we define this ratio (yield/leakage) as “gross gain” [50]. A higher value of gross gain indicates a better genetic switch with a good tightness.

Low tightness is the one problem of the G2 method. The gross gain for G2 usually ranges from several to several tens in *E. coli* [50]. This value is significantly lower than that of other established systems for tight gene expression control. For example, the *araO/araC* transcriptional regulator recorded a gross gain of over 10^3^ [57]. The low tightness is partially due to the negligible level of LTaBR that restricts the upper limit of gross gain. The LTaBR reaches 5–25% of the yield from the wild type gene without UAG-insertion in the major laboratory strains of *E. coli* [56]. A low yield and a high LTaS are also other reasons for the low tightness in some Uaa incorporation systems. These problematic points must be solved to achieve a better tightness. 

The second problem is incorporation of the Uaa in target gene products, which must alter the native sequence and may affect their function in some cases.

The third problem is that a truncated protein, which is generated by the termination of peptide elongation at the inserted UAG, is constitutively produced in the OFF-state. The truncated product may occasionally have an undesired function, such as having toxic effects.

In the following sub-sections, we describe some methods to resolve the problems of G2 for construction of G3 (Figure 5). Some factors, whose effects on switching performance are theoretically obvious, are also included, although their effects have not been experimentally confirmed.

### 5.1. Inducible UaaRS/tRNA_CUA_

An inducible UaaRS/tRNA_CUA_ is an effective method to suppress LTaS [48]. LTaS is mainly caused by a standard amino acid mischarged tRNA_CUA_. The absence of tRNA_CUA_ in the OFF-state eliminated LTaS [48,49]. In addition, LTaS was also suppressed in the absence of the UaaRS because the mischarge of tRNA_CUA_ was mostly catalyzed by the UaaRS [48]. The inducible expression of the UaaRS and/or tRNA_CUA_ is simply achieved using an inducible transcriptional element, such as *araO/araC* or *lacO/lacI* in *E. coli*.

However, the use of highly reliable genetic switching elements, whose number is limited, is not desirable when used only to improve the performance of another genetic switch. Moreover, two or more inducers are needed to control the switching. Recently, a Uaa translational switch, in which a positive feedback derepression genetic circuit was installed, has been proposed to efficiently suppress LTaS without the use of other switching elements (Figure 6) [50]. Expression of the UaaRS gene and/or tRNA_CUA_ gene was modified to be regulated in a Uaa-dependent manner by this positive feedback mechanism, such as an insertion of either UAG into the UaaRS gene or a Uaa transcriptional switch, that is described in the next section, into the tRNA_CUA_ gene. With this feedback circuit, the UaaRS and/or tRNA_CUA_ gene did not express in the absence of the Uaa, resulting in suppression of LTaS. This system is controlled only by the Uaa.

Transduction of UaaRS/tRNA_CUA_ using a viral vector is an alternative approach [58]. The transduction is useful for topical and temporal applications although only a single OFF/ON transition is possible.

### 5.2. Optimization of the UaaRS/tRNA_CUA_ Expression Level

The best efficiency and substrate specificity can be achieved by the optimization of the UaaRS/tRNA_CUA_ expression level. The accuracy of aminoacylation requires a proper balance of UaaRS and tRNA_CUA_ [59]. Over expression of UaaRS enhances the mis-acylation of tRNA_CUA_ with a standard amino acid, which is a poor substrate of the UaaRS, resulting in increased LTaS [48,60,61]. On the other hand, the efficiency and yield should be reduced when the expression level of the UaaRS/tRNA_CUA_ is too low [50]. A proper selection of promoters, ribosomal binding site, and copy number of a gene can be used to adjust the UaaRS/tRNA_CUA_ expression to an optimum level [48].

### 5.3. Multiplexed TAG Insertion in a Target Gene

LTaBR can be suppressed by a multiplexed TAG insertion in a target gene [48,50]. Incorporation of standard amino acids by host near-cognate tRNAs causes LTaBR. At the inserted UAG, the aminoacylated near cognate tRNAs compete against a peptide release factor in the OFF-state [62]. Although the Uaa-tRNA_CUA_ also similarly competes in the ON-state, the efficiency of the Uaa-tRNA_CUA_ is usually much higher. The multiplexed UAG, therefore, causes a larger attenuation of the read-through by near-cognate tRNAs than that of the Uaa-tRNA_CUA_. Finally, the gross gain increases although the yield decreases. A combination of a multiplexed UAG and a positive feedback derepression gene circuit improved the gross gain which was 3 × 10^2^-fold greater than that of the parent system although the yield decreased to 0.2-fold [50]. This method, however, is not effective if the efficiency of Uaa-tRNA_CUA_ is low and the LTaBR is not negligible because there would be considerable loss of yield.

### 5.4. Location of the UAG Insertion

The simplest method to prevent the production of truncated protein is insertion of the UAG next to the translation start codon AUG [48,49,50]. In this case, no truncated protein is produced because the first peptide elongation is not completed. However, it should be considered that the second codon/residue affects the translation efficiency, the extent of N-terminal Met excision and the half-life of the protein after the Met excision [63,64,65,66,67,68]. An insertion of the UAG at a permitted site is a general solution if the truncated protein is either not toxic or not reversed to be functional.

### 5.5. Uaa-Residue Conversion to Standard Amino Acid Residue

Use of the *N^ε^*-Acetyl-_L_-lysine (AcK) translational switch is a method to resolve the problem of the incorporated Uaa affecting the function of target gene products [69,70,71,72]. A codon(s) encoding Lys is selected to be substituted for the UAG where AcK is incorporated. AcK is an amino acid, which is frequently found in natural proteins as a product of post-translational modification [73]. The incorporated AcK is reversed to Lys by deacetylases [74,75]. The target protein in which the Lys is substituted for AcK, reverts to the unaltered protein.

### 5.6. Suppression of Peptide Release Factor

A weaker activity of peptide release factor leads to a higher yield because the efficiency is increased due to the enhanced relative activity of the Uaa-tRNA_CUA_. LTaS and LTaBR are nonetheless also increased as well as the yield, indicating that the gross gain is not expected to be further improved [56,76,77].

Some methods are reported to weaken the activity of peptide release factors. The first method is an optimization of sequence context around the UAG [72,78]. In *E. coli*, efficiency of translation termination is decreased the most if the C-terminal protein sequence contains poly-Pro stretches [79,80,81]. In addition, the first 3′ base following the UAG also affects the efficiency of UAG recognition by RF-1 [82].

The second method is an attenuated mutant of the peptide release factor [83,84]. The mutant peptide release factor with attenuated termination activity is less competitive against the Uaa-tRNA_CUA_. The efficiency of the read-through is consequently increased in a host carrying the mutant peptide release factor gene. Based on the same principle, knockout of the peptide release factor is a more effective method than the attenuated mutants. The RF-1 gene, *prfA*, was deleted in *E. coli* strains with a specific modification, such as the substitution of all or selected genomic TAG stop codons for other stop codons and the recovery of another release factor, RF-2, to the wild-type [76,85,86,87]. The efficiency reaches a maximum level in the *prfA* deletion mutants [88].

The third method is a mutant ribosome which more weakly interacts with the peptide release factor [89]. The interaction with the peptide release factor was attenuated by an appropriate modification of a nucleotide sequence in the 530 loop in 16S rRNA in *E. coli*. 

### 5.7. Uaa-Permissive Elongation Factor

The *E. coli* elongation factor EF-Tu is a component of quality control in protein synthesis [90]. A weak or too strong binding of the Uaa-tRNA_CUA_ to EF-Tu causes an inefficient incorporation of the Uaa. Some Uaa-tRNA_CUA_’s have this problem [91,92]. An engineered EF-Tu, which binds to the Uaa-tRNA_CUA_ with an appropriate affinity, can result in highly efficient Uaa incorporation.

### 5.8. Host Selection

The performance of the Uaa translational switch varies in different hosts [48,53]. For example, the level of LTaBR is widely distributed among the common laboratory strains of *E. coli*, such as 1% for TOP10 and 17% for MG1655 [56]. Selection of a suitable strain for the intended use, therefore, is important to obtain an adequate performance of the Uaa translational switch.

### 5.9. Improved UaaRS/tRNA_CUA_

Improving the performance of UaaRS/tRNA_CUA_, such as orthogonality, efficiency, and substrate specificity, is primarily important to construct a better Uaa translational switch. The most established method is a directed evolution using a positive and a negative selection in either *E. coli* or yeast (Figure 7A) [45,51,70]. To screen mutants which effectively incorporate the Uaa, positive selection is performed using an antibiotic resistant gene with an inserted TAG. In the presence of the Uaa and antibiotic, higher effective clones are selected from a library, because the antibiotic resistant gene is more strongly expressed. Following this, negative selection using a TAG-inserted toxin gene is performed. The clones with a reduced leakage translation are selected in the absence of the Uaa. The clones, which are more highly efficient and specific for the Uaa incorporation, are obtained after several rounds of the two-step selection. Nonetheless, most of the evolved UaaRSs, which were derived using this selection method, showed much reduced activity compared with that of the native enzymes [88,93,94,95,96].

Recently, some modified methods have been proposed to develop a more effective UaaRS. Effective UaaRSs can be more stringently selected using both libraries of chromosomally integrated aaRSs and a performance selection of green-fluorescent protein (GFP) fluorescence intensity [88]. In the previous method, an aaRS library was constructed in multi-copy plasmids, resulting in UaaRS/tRNA_CUA_ overexpression to overcome enzyme inefficiency. This method resolved this problem by chromosomal integration of an aaRS library. 

Another method used parallel positive selections combined with deep sequencing and statistical analysis (Figure 7B) [97]. This method does not include the negative selection step, which often deletes the most active clones from the gene pool. The parallel positive selections in the presence and absence of the Uaa can rapidly identify selectively-enriched sequences in the resulting gene pools of the Uaa-present selection using deep sequencing analysis. Finally, efficient and specific UaaRSs can be obtained.

A directed evolution by compartmentalized partnered replication has also been proposed (Figure 7C) [98]. In this method, the *E. coli* cells carrying libraries of mutant aaRS/tRNA_CUA_ genes and a TAG-inserted *Taq* DNA polymerase gene are compartmentalized by a water-in-oil emulsion. The *Taq* enzyme is selectively produced in the bacterium expressing an efficient UaaRS. The evolved UaaRS/tRNA_CUA_ is efficiently amplified when the emulsions are thermal cycled.

The phage-assisted continuous evolution is an alternative method (Figure 7D) [99]. This method links UaaRS activity to the expression of gene III, which encodes the pIII protein required for the phage to be infectious. Libraries of aaRSs are constructed in the phage. The pIII gene with an inserted UAG is encoded in a bacterial plasmid and is expressed only in the bacteria, which are infected by the phage carrying a highly active UaaRS, resulting in selective propagation of the phage. The sequences are continuously mutagenized with a defined mutational frequency. Consequently, highly-efficient UaaRSs are obtained through hundreds of generations of evolution.

One method uses a difference in the sensitivity for a natural protein degradation pathway, the N-end rule pathway (Figure 7E) [100]. The proteins containing certain desired *N*-terminal Uaa’s have longer half-lives. Using GFP as a degradation target, highly active and specific UaaRSs can be identified in cells with high GFP fluorescence emission.

In addition, molecular designing on the basis of aaRS/tRNA_CUA_ structure complements these various screening strategies (Figure 7F) [101,102,103]. Installation of an editing domain into a UaaRS is also a possible method to improve its specificity [104]. Novel methods to improve UaaRS/tRNA_CUA_s have been described in detail in other recent reviews [53,105,106].

## 6. Uaa Transcriptional Switch

The transcription of target genes can be controlled by regulation of the translation of leader peptides using a Uaa translational switch (Figure 8) [107]. This is a converter of translational-to-transcriptional regulation. A *cis*-regulatory leader-peptide element activates the transcription of downstream genes [108,109]. For example, the transcription of *trp* and *tna* operons is regulated by the translation of a leader peptide, which is controlled by the presence or absence of Trp [110,111,112,113]. Introduction of a TAG into the leader peptide region at an appropriate position enables control of the transcription of downstream target genes.

## 7. Applications

### 7.1. Biological Containment

Biological containment is a technique that genetically programs organisms to grow only in human-controlled areas, such as in the laboratory and in a factory, and to die in the natural environment [114,115]. Using this technique, it is possible to contain “useful, but dangerous” organisms, such as a genetically modified organism whose safety has not been certified, pathogens, and harmful invasive species.

A novel strategy for biological containment has been proposed that involves the Uaa translational switch as an essential genetic part (Figure 9A) [71,114,116,117]. The Uaa translational switch controls the expression of essential genes, which are either naturally existing genes in the genome or synthetic essential genes, such as antitoxin genes against co-introduced toxin genes. The organisms, in which this type of biological containment system is installed, can survive only in the presence of the Uaa.

Previous biological containment systems used a natural metabolite auxotrophy or a natural substance controlling system, which often failed to contain the organisms [118]. This failure was attributed to those natural molecules, which either occurred in extremely low amounts or were absent in the natural environment but existed unexpectedly in specific microenvironments. In contrast, the Uaa auxotrophic organisms cannot escape from the human-controlled areas because the Uaa never existed in the natural environment.

One problem is the emergence of escapers where the containment gene circuit is broken due to mutations, such as a sense mutation in the inserted TAG and a naturally occurring tRNA_CUA_. Multiplication of target genes, functionally essential Uaa residues, such as for catalysis, multimerization, and metallic chelate formation, and removal of redundant near-cognate tRNAs to prevent the generation of naturally occurring tRNA_CUA_, are possible countermeasures against the emergence of escapers [71,116,117,119,120]. 

### 7.2. Live Attenuated-Virus Vaccine

A similar technique to that of the biological containment has been applied for generating a live attenuated-virus vaccine (Figure 9B) [121,122,123]. UAGs are inserted into some genes essential for proliferation in the RNA virus genome. This vaccine strain can be amplified in the cell line, which maintains a UaaRS/tRNA_CUA_ in the presence of Uaa. Then, the virus will not proliferate in an unmodified natural host.

Generally, live vaccines more effectively elicit immune response than dead or component vaccines [124]. However, a highly pathogenic virus cannot be used as a live vaccine because it could cause severe illness. This is why a virus, whose pathogenicity is attenuated, is used as a vaccine strain. One problem is that immune induction is reduced if the virus is too attenuated. In addition, the traditional attenuated virus is sometimes pathogenic for sensitive individuals and possibly reverts to a highly pathogenic form. The Uaa-dependent live attenuated-virus vaccine strongly induces immunity and provides protection against subsequent infection. Moreover, the vaccine strain rarely diffuses out and is not maintained within an open environment.

### 7.3. High-yield and Zero-Leakage Expression System (HYZEL)

HYZEL is constructed with a Uaa translational switch in combination with a transcriptional regulatory system (Figure 9C) [48]. In this system, the production of target gene proteins is tightly controlled by the transcription-translation dual regulation [48,72]. Both transcription and translation are shut-down to completely eliminate the leakage expression in the OFF-state. On the other hand, the maximum expression is induced under the full activation of both transcription and translation in the ON-state.

HYZEL is useful for production of highly toxic proteins. In microbial production, toxin expression is strictly suppressed during the proliferation stage because a slight leakage expression will kill the host cells. Using HYZEL, the host microbes can proliferate without affecting fitness because of complete suppression of toxin leakage expression. The maximum yield can be achieved by induction after sufficient proliferation. A HYZEL system using a combination of the 3-iodo-_L_-tyrosine incorporation system, *P_BAD_*-araO/araC regulated T7 RNA polymerase, and *P_T7_-lacO/lacI* regulated target gene, maintained an expression construct for the potent toxin colicin E3, which kills its host *E. coli* with a few molecules, in a multicopy plasmid carrying pBR322 replication origin [48]. Other applications of HYZEL for fine tuning of gene expression could be beneficial. For example, controlling the expression level of enzymes in metabolic engineering may be a promising strategy.

## 8. Conclusions

Conditional UAG read-through, which was identified half a century ago, is one of the most established methods for controlling the translation of target genes and is still in use today. Site-specific Uaa incorporation using an expanded genetic code enables a UAG read-through controlled by the Uaa which is a bio-orthogonal and unnatural substance. Although site-specific Uaa incorporation was not originally developed for switching use, the Uaa translational switch has now been optimized by various techniques, such as an inducible UaaRS/tRNA_CUA_ [48,50,58], an adjustment of the UaaRS/tRNA_CUA_ expression level [48,50], multiplexed the UAG insertion in a target gene [48,50], selection of the UAG insertion position [48,50,72], a Uaa-residue converting to a standard amino acid residue [69,70,71,72], optimization of sequence context around the UAG [72,78], modification or deletion in a peptide release factor [76,83,84,85,86,87,88], modifications of elongation factor and the ribosome [89,91,92], and use of a further evolved UaaRS/tRNA_CUA_ with a higher efficiency and specificity [88,97,98,99,100,101,102,103,104]. 

In the Michael Crichton’s novel “Jurassic Park”, the dinosaurs were contained using a natural amino acid Lys auxotrophy [114,125]. However, they escaped from the human controlling area by learning that eating Lys-rich foods existing in the natural environment allowed them to survive. Now, a biological containment using Uaa auxotrophy has been realized by the Uaa translational switch. This containment strategy should open the door to safe use for genetically modified microorganisms in an open environment, such as for biological remediation, vaccine production, and microbial formulation. The advanced Uaa translational switch is also useful for tight and fine tuning of gene expression during highly toxic protein production and metabolic engineering.

The “genome write” projects are progressing for some microbes [126,127,128,129]. The codon recoding is one of the goals in the early stages. The codon recoding should generate some blank codons, which can be assigned to the Uaa’s. An orthogonal ribosome which efficiently reads quadruplet codons has also been created [130]. At the present time, the Uaa translational switches mostly use only UAG. The utility of Uaa translational switches may be dramatically increased by expansion of the range of target codons which will be generated by the codon recoding.

## Figures and Tables

**Figure 1 ijms-20-00887-f001:**
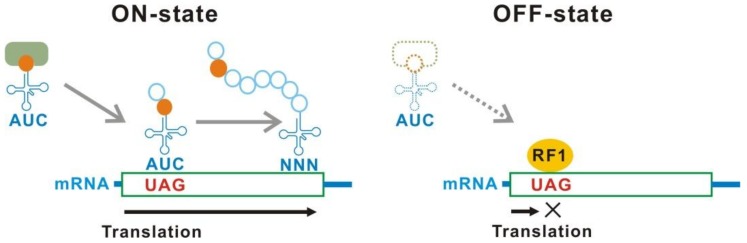
Translational switch using a UAG read-through in *Escherichia coli*. In the ON-state, an aminoacyl-tRNA_CUA_ incorporates the amino acid at an inserted UAG in a target mRNA, resulting in a stop codon read-through and full-length translation of the target gene products. On the other hand, when the aminoacyl-tRNA_CUA_ is not supplied in the OFF-state, the UAG is solely recognized as a stop codon and the synthesis of full-length products is interrupted by the peptide release factor 1. Circles and boxes indicate amino acids and protein coding regions in mRNAs, respectively. aaRS, aminoacyl-tRNA synthetase. tRNA_CUA_, UAG suppressor tRNA. RF1, peptide release factor 1.

**Figure 2 ijms-20-00887-f002:**
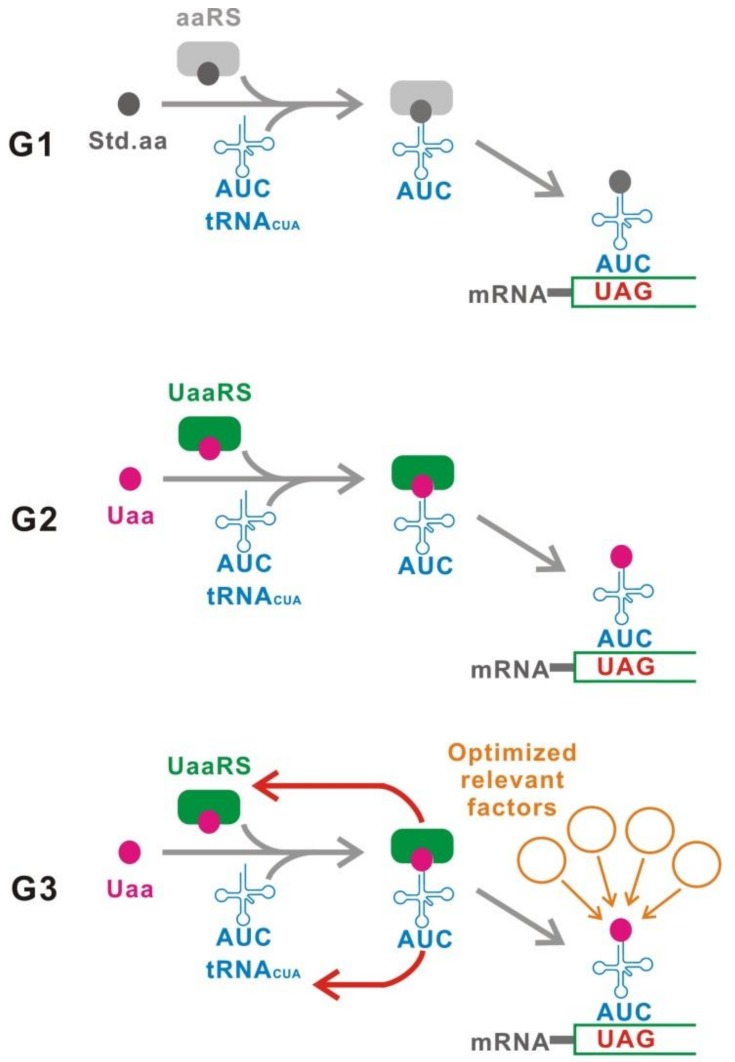
Classification. The first generation (G1) uses a tRNA_CUA_ carrying a standard natural amino acid. In contrast, an unnatural aminoacyl-tRNA synthetase/tRNA_CUA_ pair suppresses UAGs in the second generation (G2). The third generation (G3) is an upgraded G2 that achieves the best performance for switching. Figure 4 illustrates G3 in detail. Std.aa, a standard amino acid. Uaa, an unnatural amino acid. UaaRS, unnatural aminoacyl-tRNA synthetase.

**Figure 3 ijms-20-00887-f003:**
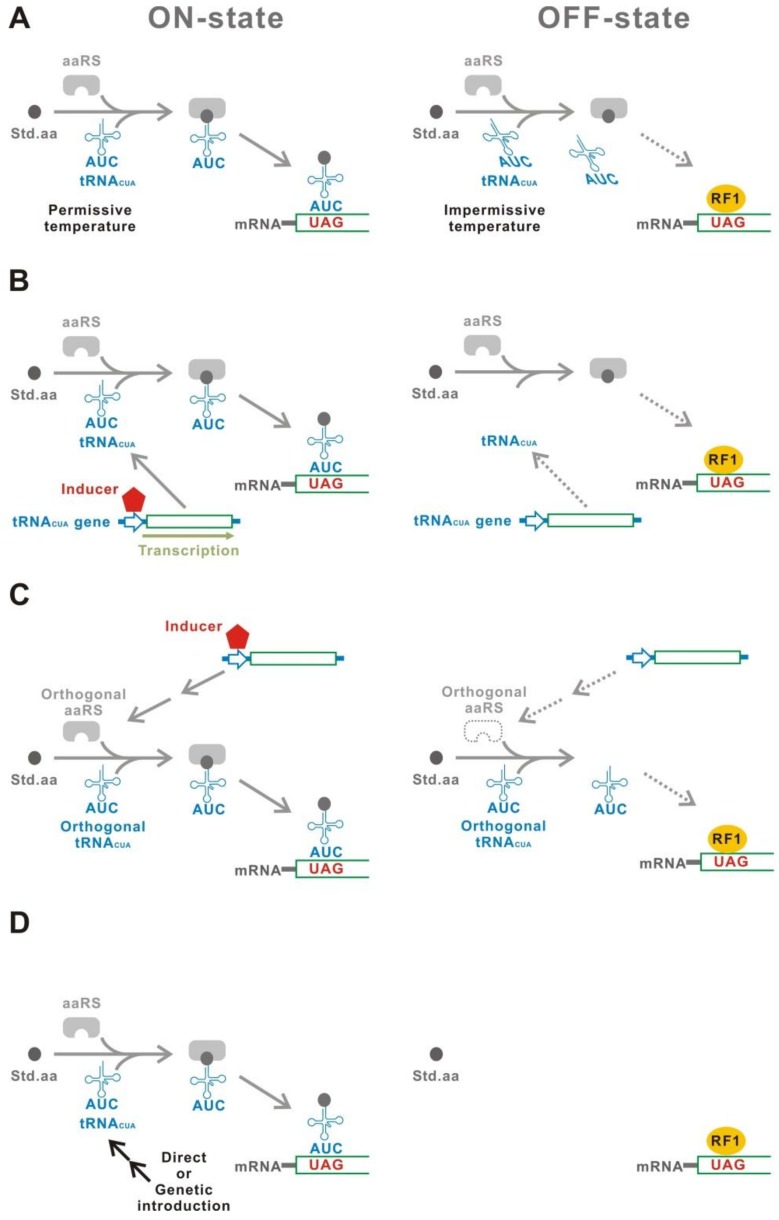
G1. (**A**) Thermolabile tRNA_CUA_. The tRNA_CUA_ is functional only at the permissive temperature. (**B**) Inducible transcription of tRNA_CUA_. The tRNA_CUA_ gene is expressed only in the presence of an inducer. (**C**) Inducible aaRS. The aaRS is produced only in the presence of an inducer. The aaRS/tRNA_CUA_ pair must be orthogonal to those of a host organism. (**D**) Introduction of the tRNA_CUA_ gene. The tRNA_CUA_ gene can be introduced using either a physical method such as electroporation or a genetic method such as crossing between a tRNA_CUA_ gene carrying a strain and a target gene carrying a strain. Open arrows indicate the promoters.

**Figure 4 ijms-20-00887-f004:**
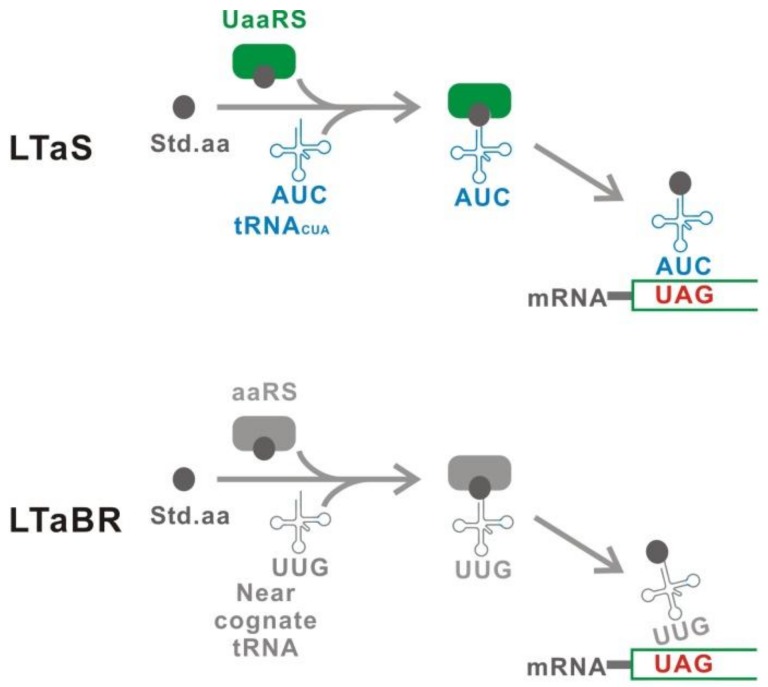
Two components of the leakage translation. The leakage translation attributed to the switch (LTaS) is mainly caused by misacylation of tRNA_CUA_ with a natural amino acid, which is a poor substrate of UaaRS. The leakage translation attributed to the basal read-through (LTaBR) is derived from read-through caused by near-cognate tRNAs. The LTaBR is independent of the performance of the Uaa translational switch.

**Figure 5 ijms-20-00887-f005:**
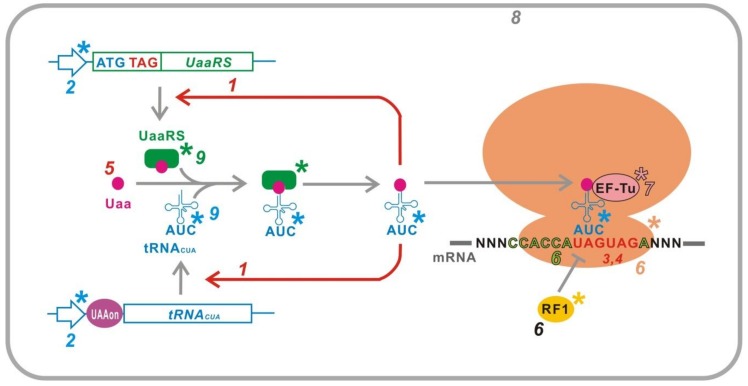
G3. Various relevant factors for switching are optimized in G3, such as (1) inducible UaaRS/tRNA_CUA_, (2) optimization of UaaRS/tRNA_CUA_ expression level, (3) multiplexed UAG insertion in a target gene, (*4*) location of the UAG insertion, (5) Uaa-residue converting to a standard amino acid residue, (*6*) suppression of peptide release factor, (*7*) Uaa-permissive elongation factor, (*8*) host selection, and (9) improved UaaRS/tRNA_CUA_. Asterisk indicates a mutant optimized for switching use. UAAon, a transcriptional ON-switch controlled by Uaa.

**Figure 6 ijms-20-00887-f006:**
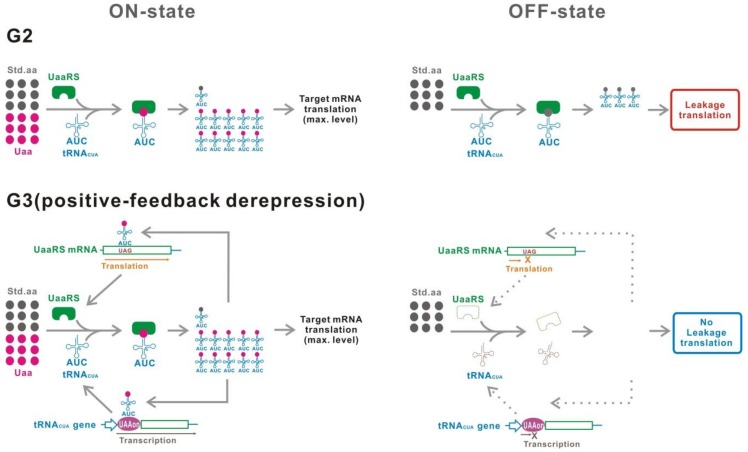
Tight G3 translational switch using a positive-feedback derepression gene circuit. In G2, the UaaRS and tRNA_CUA_ are expressed even in the OFF-state, causing a significant level of leakage translation. A positive-feedback derepression circuit from the Uaa-tRNA_CUA_ to the UaaRS and/or tRNA_CUA_ gene decreases the expression level of those genes in the OFF-state, resulting in a suppression of leakage translation.

**Figure 7 ijms-20-00887-f007:**
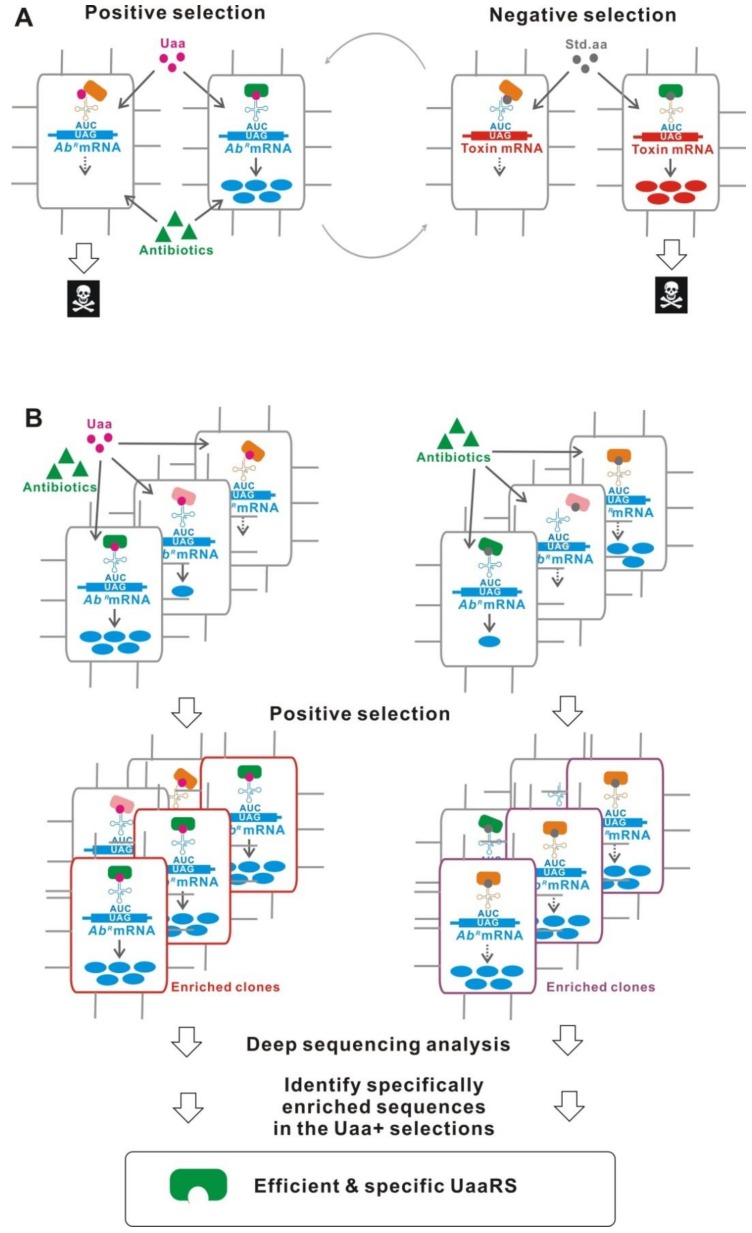
Improvement of UaaRS/tRNA_CUA_. (**A**) Traditional evolution method. A positive selection and a negative selection are performed to effectively isolate the Uaa incorporating and less Std. aa incorporating UaaRSs, respectively. **(B**) Parallel positive selections combined with deep sequencing and statistical analysis. Positive selections are performed in the presence and absence of Uaa. Specifically-enriched sequences in the Uaa selection, which encode efficient and specific UaaRSs, are identified after deep sequencing. (**C**) Compartmentalized partnered replication. DNA sequences that encode highly active UaaRSs are selectively amplified in compartmentalized micro-drops. (**D**) Phage-assisted continuous evolution. Phage carrying a highly active UaaRS are selectively propagated. (**E**) Post-translational proofreading. The proteins containing certain desired *N*-terminal Uaas have longer half-lives. Genes encoding highly specific UaaRSs are identified in bacteria expressing high amounts of GFP. The bacteria can then be isolated using a cell sorter. (F) Molecular design based on structural data.

**Figure 8 ijms-20-00887-f008:**
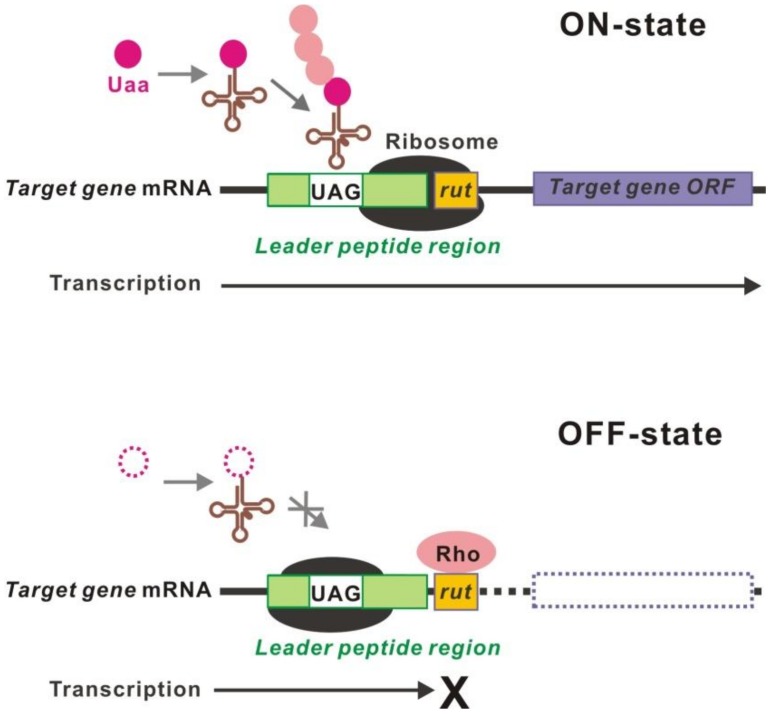
Uaa transcriptional switch. The transcription of the target gene open reading frame (ORF) requires the translation of a leader peptide beyond the inserted UAG. In the absence of the Uaa, translation of a leader peptide is interrupted, resulting in transcriptional termination at the unmasked *rut* site.

**Figure 9 ijms-20-00887-f009:**
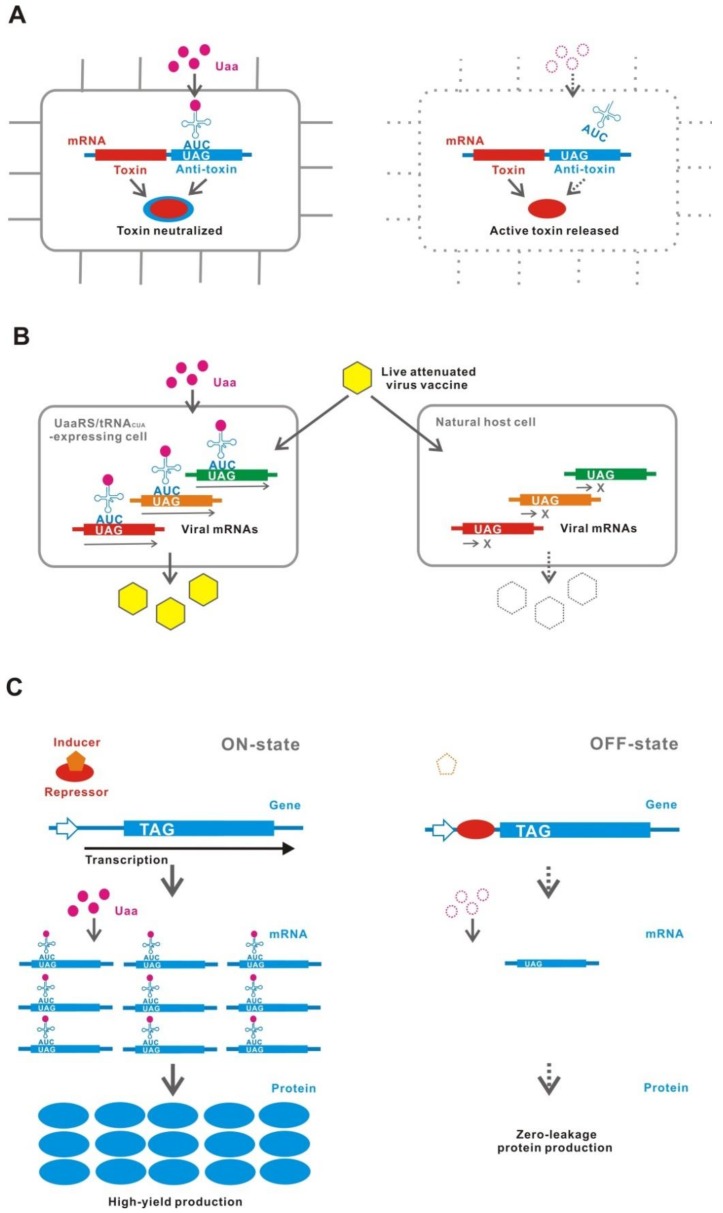
Applications. **(A**) Biological containment. An active containment system using a toxin-antitoxin gene is shown. Bacteria survive only in the presence of the Uaa because the antitoxin gene has a TAG insertion and is expressed in a Uaa-dependent manner. (**B**) Live attenuated-virus vaccine. The virus is propagated only in a cell-line expressing UaaRS/tRNA_CUA_ in the presence of the Uaa but is not propagated in natural host cells. (**C**) High-yield and zero-leakage expression system. In the usual transcriptional controlling system, a small amount of mRNA is produced even in the OFF-state, resulting in a leakage (production) of protein. A combination with translation control using a Uaa translational switch can completely suppress the translation of the mRNA in the OFF-state.

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
