# Peer review of "Translational Control using an Expanded Genetic Code"

_ijms, 2019, doi:10.3390/ijms20040887_

Round 1
Reviewer 1 Report
This article by Kato is an interesting review that summarizes the history and recent advance of the translational control by conditional read-through at internal UAG stop codon and essentially focuses on methods for site-specific unnatural amino acid (Uaa) incorporation. These approaches can have a wide range of applications, such as Uaa-auxotrophy-based biological containment, live-attenuated vaccine and high-yield expressions systems. The review will certainly be of interest to scientists from the synthetic biology community.
The review is reasonably well written but the English would still require to be edited. Nevertheless, I would recommend addition of few figures. For instance, it would be great to have a figure for illustrating some of the methods used to improve the performance of UaaRS/tRNAcua (paragraph 5.9). Additionally, it would also be necessary to explain better some of these methods. For instance, it is not well explained how these methods of selection work. It would also be great to have a figure for the paragraph "6. Uaa transcriptional switch".
Finally, a figure illustrating the different applications would also be helpful.
Additional minor comments
P.3 lanes 85 to 88:
This sentence is not clear at the level of the underlined section.
"G1 is widely used for functional analysis of either a natural TAG mutant genes or a synthetic library of TAG-insertion mutant genes [9,10]."
This sentence is not understandable.
"The minimal alteration of a target gene sequence is an 87 advantage of the translational switches including G1 and the following generation methods."
P.4 lanes 136 to 139:
The sentence starting "in the early 2000's..." is not clearly written.
P.5, lanes 161-163:
This sentence should be rewritten:
"However, this system was originally developed to alter target proteins, such as for structural analysis, labeling, and functional modification of proteins [51-53]. "
Proposal:
"However, this system was originally developed to alter target proteins for various purposes including structural analysis, labeling, and functional modification of proteins [51-53]. "
P.5, lanes 177-180:
It would be nice to underline the first letters of:
" ... Leakage Translation attributable to the Uaa translational Switch (LTaS)." and
"... Leakage Translation attributable to the Basal Read-through (LTaBR)."
P5. lanes 192-193:
This sentence should be rewritten:
"The low tightness is partially because the value of LTaBR 192 restricts the upper limit of gross gain."
P7. lane 229.
"elements" instead "elemments".
P7. lanes 229- 231.
This sentence is not clear and should be rewritten:
""The expression of UaaRS and/or tRNACUA was altered to be dependent on the Uaa by this positive feedback mechanism, such as an insertion of either UAG into the UaaRS gene or a Uaa transcriptional switch that was mentioned in the next section. "
P8. lanes 284- 285.
This sentence is not clear and should be rewritten:
"Similar to this, a more effective method is knockout of the 284 peptide release factor. "
P9. lane 306.
This sentence is not clear and should be rewritten:
The positive selection uses an antibiotic resistant gene with the TAG inserted.
P11. First paragraph (lanes 407-417).
It would be good to cite here the references corresponding to the various techniques mentioned in the text.
Author Response
First, we thank reviewer 1 for the useful comments for improving our manuscript. Our responses to the specific comments are detailed below.
The reviewer’s comments are indicated in black.
Our responses are written in blue.
Changes in the revised manuscript are highlighted in green.
The review is reasonably well written but the English would still require to be edited.
Our revised manuscript has been checked by a native English proofreader prior to submission.
Nevertheless, I would recommend addition of few figures. For instance, it would be great to have a figure for illustrating some of the methods used to improve the performance of UaaRS/tRNAcua (paragraph 5.9). Additionally, it would also be necessary to explain better some of these methods. For instance, it is not well explained how these methods of selection work. It would also be great to have a figure for the paragraph "6. Uaa transcriptional switch".
Finally, a figure illustrating the different applications would also be helpful.
We agree with the reviewer’s suggestion. We have added some figures (Graphical abstruct, figures 3 and 6-9) in our revised manuscript. Figures 3 and 4 in the original manuscript have been renumbered as Figures 4 and 5, respectively). In addition, we have added some information on the methods for the selection in paragraph 5.9 (p.10, line 341-343; p.13, line 368; p.13, line 372-376 in our revised manuscript).
Additional minor comments
P.3 lanes 85 to 88:
This sentence is not clear at the level of the underlined section.
"G1 is widely used for functional analysis of either a natural TAG mutant genes or a synthetic library of TAG-insertion mutant genes [9,10]."
This sentence is not understandable.
"The minimal alteration of a target gene sequence is an advantage of the translational switches including G1 and the following generation methods."
We have rewritten these sentences as follows.
“G1 is widely used for functional analysis of specific genes by conditional rescue of either a naturally occurring or a synthetic TAG-insertion into mutant genes [9,10].” (p.2, line 81-82 in our revised manuscript)
“The TAG-insertion in a target gene sequence may be technically simple and easy, suggesting that this is an advantage of the translational switches, including both G1 and the following generation methods.” (p.2, line 82-84 in our revised manuscript)
P.4 lanes 136 to 139:
The sentence starting "in the early 2000's..." is not clearly written.
We have rewritten this sentence as follows.
“In the early 2000’s, an orthogonal tRNACUA was successfully aminoacylated using a Uaa-specific mutant of aaRS (UaaRS), enabling use of this method in vivo. [42-47].” (p.5, line 147-149 in our revised manuscript)
P.5, lanes 161-163:
This sentence should be rewritten:
"However, this system was originally developed to alter target proteins, such as for structural analysis, labeling, and functional modification of proteins [51-53]. "
Proposal:
"However, this system was originally developed to alter target proteins for various purposes including structural analysis, labeling, and functional modification of proteins [51-53]. "
We have replaced this sentence with the proposed sentence (p.6, line 172-174 in our revised manuscript). We thank the reviewer for this suggestion.
P.5, lanes 177-180:
It would be nice to underline the first letters of:
" ... Leakage Translation attributable to the Uaa translational Switch (LTaS)." and
"... Leakage Translation attributable to the Basal Read-through (LTaBR)."
We have modified this portion as the reviewer has suggested (p.6, line 188 and 191-192 in our revised manuscript).
P5. lanes 192-193:
This sentence should be rewritten:
"The low tightness is partially because the value of LTaBR restricts the upper limit of gross gain."
We have rewritten this sentence as follows.
“The low tightness is partially due to the negligible level of LTaBR that restricts the upper limit of gross gain.” (p.6, line 203-204 in our revised manuscript)
P7. lane 229.
"elements" instead "elemments".
We have corrected this word as the reviewer has suggested (p.7, line 233 in our revised manuscript).
P7. lanes 229- 231.
This sentence is not clear and should be rewritten:
""The expression of UaaRS and/or tRNACUA was altered to be dependent on the Uaa by this positive feedback mechanism, such as an insertion of either UAG into the UaaRS gene or a Uaa transcriptional switch that was mentioned in the next section. "
We have rewritten this sentence as follows.
“Expression of the UaaRS gene and/or tRNACUA gene was modified to be regulated in a Uaa-dependent manner by this positive feedback mechanism, such as an insertion of either UAG into the UaaRS gene or a Uaa transcriptional switch, that is described in the next section, into the tRNACUA gene.” (p.7, line 233-236 in our revised manuscript)
P8. lanes 284- 285.
This sentence is not clear and should be rewritten:
"Similar to this, a more effective method is knockout of the peptide release factor."
We have rewritten this sentence as follows.
“Based on the same principle, knockout of the peptide release factor is a more effective method than the attenuated mutants.” (p.10, line 301-302 in our revised manuscript)
P9. lane 306.
This sentence is not clear and should be rewritten:
The positive selection uses an antibiotic resistant gene with the TAG inserted.
We have rewritten this sentence as follows.
“To screen mutants which effectively incorporate the Uaa, positive selection is performed using an antibiotic resistant gene with an inserted TAG.” (p.10, line 323-324 in our revised manuscript)
P11. First paragraph (lanes 407-417).
It would be good to cite here the references corresponding to the various techniques mentioned in the text.
We have cited those references in this paragraph (p.16, line 466-472 in our revised manuscript).

Reviewer 2 Report
This review nicely reviewed recently development of genetic code expansion for Uaa incorporation. It listed many subtitles which is good for reading. I only have two minor comments:
1) Figure 3, LTaBR model used near cognate tRNA (UUA) as an example. But it is actually an suppressor tRNA for TAA stop codon. So it could be better to use sense codon near cognate tRNA such as tRNAGln (UUG) as a representative.
2) Section 6 is some kinds of away from the genetic code expansion which could be removed.
Author Response
First, we thank reviewer 2 for the useful comments that have helped to improve our manuscript. Our responses to the specific comments are detailed below.
The reviewer’s comments are indicated in black.
Our responses are written in blue.
Changes in the revised manuscript are highlighted in green.
1) Figure 3, LTaBR model used near cognate tRNA (UUA) as an example. But it is actually an suppressor tRNA for TAA stop codon. So it could be better to use sense codon near cognate tRNA such as tRNAGln (UUG) as a representative.
We have replaced the “UUA” with “UUG” as the reviewer has suggested (Figure 4 in our revised manuscript).
2) Section 6 is some kinds of away from the genetic code expansion which could be removed.
We feel that the technique described in Section 6 is an important application of the Uaa amber suppression. Although we agree that the reviewer’s suggestion is a possible alternative, we have kept this section in our revised manuscript.
